# Getting it right: suppression and leveraging of noise in robust decision-making

Rishikesh P. Bhalerao

Department of Forest Genetics and Plant Physiology, The Swedish University of Agricultural Sciences, Umeå Plant Science Center, Umeå, Sweden

Noise; stochastic resonance; robustness.

**Corresponding author:**
Rishikesh P. Bhalerao
Email: rishi.bhalerao@slu.se

**Associate Editor:**
Iain Johnston

### Abstract

Noise is a ubiquitous feature for all organisms growing in nature. Noise (defined here as stochastic variation) in the availability of nutrients, water and light profoundly impacts their growth and development. Not only is noise present as an external factor but cellular processes themselves are noisy. Therefore, it is remarkable that organisms can display robust control of growth and development despite noise. To survive, various mechanisms to suppress noise have evolved. However, it is also becoming apparent that noise is not just a nuisance that organisms must suppress but can be beneficial as low noise can facilitate the response of an organism to a sub-threshold input signal in a stochastic resonance mechanism. This review discusses mechanisms capable of noise suppression or noise leveraging that might play a significant role in robust temporal regulation of an organism's response to their noisy environment.

## Introduction

All organisms display remarkable robustness in outputs, such as the timing of developmental transitions, which are often controlled by signals (external or internal) that are inherently noisy, i.e., display considerable stochastic variation both temporally and spatially. For example, temperature may be considered to vary daily or hourly (with changes typically in the range of 4–25°C over 24 hours) (Chris, 2023; Ma et al., 2015). Similarly, available light may vary between 100 and 1500 PPFD during the course of the day and may fluctuate hourly due to changes in cloud cover (Sanchez et al., 2015). Moreover, water and nutrient availability may vary spatio-temporally. In addition to such daily/hourly variations, seasonal variations occur on longer timescales, with winter–summer temperatures varying by up to 50 °C. While external sources of noise (external noise) are obvious, variations that occur within an organism at the cellular or tissue level, i.e. internal noise, are equally important. For example, all steps from signal detection to final output, i.e., transduction, contribute to noise, with not only ligand concentrations varying but also binding of ligands to its receptor, gene expression (including stages of transcription, e.g., promoter binding by transcription factors) and translation all display stochastic variation. Hence, every binding event of a ligand to a receptor may not lead to activation of downstream output (Suderman et al., 2017; Turner et al., 2010). Similarly, every event of binding of a transcription factor to a promoter may not lead to activation of transcription (Krn et al., 2005). For example, experiments on *Escherichia coli* have demonstrated variations in gene expression within a single cell as well as between cells that could be as high as 5 fold, highlighting noise from transcription (Elowitz et al., 2002). Similar observations of transcriptional noise have also been made in plants (Araújo et al., 2017; Cortijo et al., 2019). What could be the source of this 'internal noise'? There may be many causes, but among them are the non-uniform distribution of molecules within a cell and the sheer probabilistic nature of biochemical reactions, especially when the involved molecules are present at low levels (McAdams & Arkin, 1999). Whatever the cause of internal noise, the consequence is that both the detection and transmission of information are also noisy. Thus, a cell or organism operates with far less than 100% efficiency in detecting and processing information.

In addition to 'within' cell variation, cell–cell variation can be observed within the same tissue. For example, in *Arabidopsis sepals*, growth rates at the tip and lateral side differ by up to 4 fold (Hervieux et al., 2016), and the timing of the cell cycle varies by almost 5-fold (12 to 60 hours) (Roeder et al., 2010). While such regional, spatial variation between tip and sepal

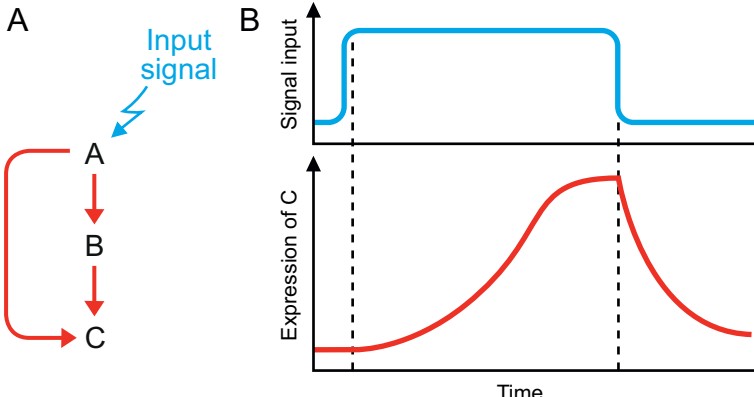

**Figure 1.** A coherent feedforward loop-mediated noise suppression.
(A) A schematic genetic network for the activation of C in response to the input signal. The expression of C requires both A and B. (B) Upon perception of the input signal (top panel, blue line), transcription factor A is activated , which then activates its downstream target B. A and B together then activate C. Since both A AND B are required for the activation of C, there is a delay in the activation of the expression of C (red line) after the signal is perceived. However, when the signal is switched off, C is downregulated rapidly. Thus , this coherent feedforward loop permits activation of the pathway only when signals persist thereby suppressing response to transient or spurious signals.

cannot be considered as true noise as it is not random, i.e. initially tip side always grows faster than the lateral side in sepals. It is however essential to note that not all cells at the tip or on the lateral side of the sepal grow at the same rate , and in fact, considerable variability in growth is observed between adjacent cells in the sepal, and variation in timing of cell cycle is also spatially random. Thus , these examples illustrate that even at the tissue or organ level, there is spatial variability and noise, and importantly, experimental data shows that such variability is important for robust development (Hong et al., 2016).

Despite external and internal noise, organisms display a remarkable ability to extract relevant information and maintain robustness (Stelling et al., 2004). Therefore, a fundamental question that needs to be answered is how organisms make robust decisions despite external and internal noise. Before addressing this, it is worth noting that although all stochastic variations can be considered noise, it is important to distinguish between noise that is random and without any inherent structure, e.g., a sudden change in available light due to cloud cover, and variability e.g., temperature variation occurring daily within a certain range or over a longer timescale during different seasons, which is strictly speaking not noise (as temperature during the day is almost always warmer than at night or temperature is always cold in winter than in summer). Presumably, dealing with noise as opposed to somewhat predictable variability over longer scales may entail distinct mechanisms.

Although attention has focused on how organisms suppress noise to make robust decisions (Balázsi et al., 2011), there is increasing evidence that noise is not just a nuisance but can counterintuitively play a crucial role in robust decision-making (A. H. Roeder, 2018; Topham et al., 2017). This also raises the question of how organisms measure or extract relevant information when faced with noisy inputs. It is highly interesting to study noise and how it shapes plant growth as plants are sessile, face highly variable, i.e. noisy environments, and lack a central organiser such as the brain. This review briefly addresses the importance of noise suppression but also focuses on how noise can be exploited to make robust decisions and the ways by which genetic networks can encode this. Whereas various roles of noise and its origins have been intensively studied in non-plant models, this topic is also increasingly studied in plants (Araújo et al., 2017; Cortijo & Locke, 2020; Wu et al., 2022) and several reviews on this topic have been published (Abley et al., 2024; Cortijo & Locke, 2020;

A. H. Roeder, 2018; Long & Boudaoud, 2019). Also, the role of noise in development, especially in the context of spatial regulation, has been addressed in plants (Adrian et al., 2015; Hong et al., 2016; Meyer et al., 2017). However, noise suppression or its leveraging in temporal context is still not as well explored in general as in plants and is the main focus of this review.

## Noise nuisance and its suppression

Given that under natural conditions, multiple sources of noise can compromise survival, it is fascinating that organisms nevertheless display highly robust responses to noisy inputs. This has prompted research into mechanisms by which cells and organisms distinguish noise from relevant input. A plethora of mechanisms enabling organisms to cope with noise have been identified that operate from timescales of seconds to minutes and even longer. One mechanism is genetic redundancies (multigene families) and overlapping pathways that provide a mode of noise reduction (Kafri et al., 2006). However, this is largely a generic mechanism for noise reduction, and in evolution, such redundancies are usually transitional (Lynch & Conery, 2000) rather than conserved over long timescales, although there are exceptions (Nowak et al., 1997).

In contrast with generic mechanisms, such as redundancy, other mechanisms involve suppression of noise or buffering, e.g., feedback loops, which enable high fidelity and robustness in behaviour (Balázsi et al., 2011; Stelling et al., 2004). Interestingly, the study of electrical circuits has revealed engineering principles and highlighted the role of feedback that enables robust output when amplifiers are faced with noisy signals (Black, 1934). Since then, in biology, several studies (Alon, 2007) have shown how noise suppression can be encoded genetically, providing an analogy to electrical circuits where such feedback was identified (Khammash, 2016). A few of these mechanisms are discussed below, including both genetic and non-genetic noise coping mechanisms (see Balázsi et al., 2011) for extensive discussion of this aspect). The examples were mainly selected because they also incorporate a temporal aspect of noise suppression. This is particularly relevant as robustness in the timing of a response in many biological processes, particularly adaptive responses, is central to survival and fitness.

A classic example of a genetically encoded noise suppression mechanism is the C1 type feed-forward loop with sign-sensitive delay (Figure 1A). Sign-sensitive delay is defined as a circuit that

functions in such a way that it facilitates a rapid response to step-like stimuli in one direction (e.g. ON to OFF) whereas response is delayed to steps in the opposite direction (OFF to ON) (Figure 1B). Such a circuit acts as a persistence detector and guards against activation of a pathway by spurious, noisy input (Mangan et al., 2003). As shown in Figure 1A, the ultimate output of the pathway in response to an input signal is the activation of transcription of C. The pathway operates with an 'AND' logic gate at the promoter of C, i.e., it requires binding of both its upstream activators A and B. Since both A AND B are required for the activation of C, there is a delay in the activation of C after the signal is perceived. This delay in turn depends upon a threshold for activating C (and in principle of A and B) dictated by various biochemical parameters, e.g., how fast A is made and switched off. Additionally important is the affinity strength of A binding to the promoter of B and A and B binding to the promoter of C; the lower the affinity, the longer the delay. In contrast, if the signal input stops, there is almost rapid switching off of the pathway. This is because activation of C requires both A and B and loss of signal rapidly leads to switching off of A synthesis. As a consequence of the delay after sensing the input signal and rapid switching off after the loss of input, this type of network can protect against spurious/noisy or fluctuating/transient inputs. This mechanism has been observed in the arabinose operon (Mangan et al, 2003). Genetic and modelling studies have shown that this circuit operates with a delay of ca. 20 mins for activation of the pathway in response to cAMP in *E. coli* and can therefore facilitate coping with spurious fluctuations of cAMP encountered by *E. coli* in nature. Thus, this example illustrates how noise suppression is genetically encoded and presumably evolved to account for noisy input aligned with temporal aspects of noise encountered under natural conditions. Although one example was highlighted, there are many examples of this type of network topology (Kittisopikul & Süel, 2010; Mangan & Alon, 2003).

Another interesting mechanism of noise suppression, but one that is non-genomic is the dynamic response of the Dionaea muscipula (Venus flytrap) in capturing prey. Closure of the trap is activated when hovering prey is sensed, brushing against its sensory hairs. Trap closure is an energy-requiring process. Thus, the plant needs to guard against spurious or random brushing of the sensory hairs produced by only fleeting encounters with the prey or unrelated effects. Trap shutting is activated by the firing of an activation potential in sensory hair cells. Work by Hedrich and colleagues (Böhm et al., 2016; Escalante-Pérez et al., 2011) showed that trap closure is not activated unless the activation potential in the hairs is fired repeatedly, at least 2–4 times within 10–20 seconds, by an insect brushing against the sensory hairs. Moreover, a crucial feature of this mechanism is that each firing of the activation potential is sub-threshold, and therefore, only repeated stimulation within the short timeframe (10–20 s) can increase the action potential above the threshold level, leading to rapid closure of the trap within a second. The reason for rapid, repeated stimuli resulting in action potential reaching above the threshold is that when stimulation events occur repeatedly (and rapidly), action potential from one event does not fully decay, before the next firing. Thus, in contrast to when an insect hovers around the hairs of the trap and brushes against the trap hairs repeatedly, a random or fleeting encounter with the prey will not cause the trap to shut (as action potential has already decayed before it is fired again). Again, this example shows how a mechanism can protect against spurious or noisy input and incorporates a temporal aspect in regulation. In this particular case, the mechanism involves a high threshold for activation of trap closure, which delays the response to counter random noise.

The last two examples highlight how noise suppression in connection with the temporal regulation of a response can be achieved. In the case of the *E. coli* arabinose operon, the mechanism can cope with spurious spikes of cAMP that could be encountered under natural conditions within a period of about 20 minutes, whereas in the Venus flytrap, noise suppression is on a timescale of a few seconds to avoid random encounters with the prey. Thus, these examples show how noise suppression can be embedded (genetically or electrically) in a temporal context of response regulation. Similar principles of noise suppression can be envisioned for responses to spatio-temporal variations in nutrient variability, light, or other external factors. Such mechanisms are particularly important in plants because they cannot move. Moreover, the basic principles may function over longer timescales, e.g., in plants, many crucial developmental decisions involve the integration of noisy signals over very long timescales of days or even months e.g. vernalisation or tree phenology (Antoniou-Kourounioti et al., 2021; Zhang et al., 2023).

The above examples indicate how organisms may avoid responding to spurious pulses or noise to achieve a robust response. However, other mechanisms have also been demonstrated to buffer noise such as negative feedback loops (to maintain homeostasis via repressors) or negative autoregulation (see (Alon, 2007) for details) (Becskei & Serrano, 2000) as well as diffusion-based dilution, which offer a way in a multicellular context to reduce cell-cell variability of signals across a group of cells (Benzi et al., 1982). Additionally, in Arabidopsis, recent evidence for example suggests that transcriptional noise can be buffered by translational control (Wu et al., 2022). Whereas the response delay of the arabinose operon shows how such mechanisms can be genetically encoded, other examples, such as the Venus flytrap shutting, demonstrate that non-genomic mechanisms are also utilised for noise suppression.

An important feature of the mechanisms above, and especially ones that have a temporal context, is that a threshold sets the bar for activation and guards against noise by delaying the response. However, this raises the question of memory (or the storage of information) that needs to be balanced against activation by noise. For example, in the Venus flytrap, once the sub-threshold potential has been activated, if the decay is rapid, the system would be tighter but with a high incidence of false negatives and loss of potential prey. Thus, the threshold for firing the activation potential must be sufficiently high to protect against random spikes but low enough so that repeated activation can push the signal beyond the threshold. In this example, the deactivation rate can be considered a type of information storage, but it is not always clear how memory is encoded, and this aspect needs to be addressed if noise suppression is to be understood fully.

## Leveraging noise benefits

While the above examples highlight the troublesome aspects of noise and how a response to it could be suppressed or buffered, there are increasing examples of how noise can be leveraged, i.e., have a beneficial role. In contrast with DNA polymerase, in retroviruses, a lack of proofreading activity in reverse transcriptase (Battula & Loeb, 1974) results in a relatively high variability in the viral DNA. This 'noise' is beneficial to the virus as some variants may be able to escape antiviral treatments (Meissner et al., 2022; Preston et al., 1988). Similar population-scale leveraging of noise can be found in bacteria (Lopez et al., 2009). For example, internal noise in *Bacillus subtilis* results in a highly variable population of otherwise genetically identical cells that differ in their ability to uptake DNA during stress when the majority of cells sporulate.

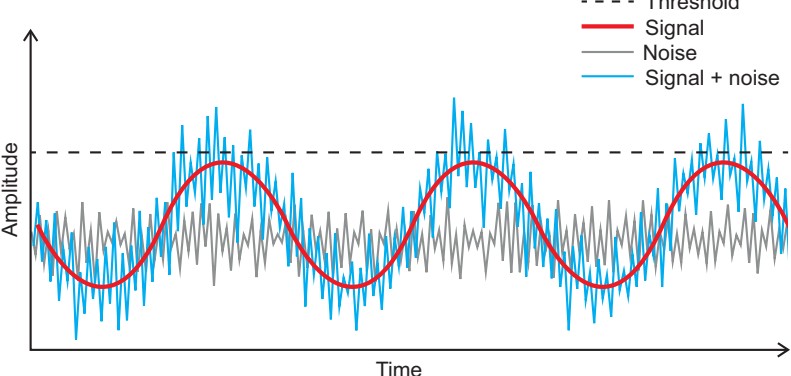

**Figure 2.** A schematic description of the stochastic resonance mechanism.
The red line indicates a periodically fluctuating signal, light grey represents stochastic or random noise (noise without inherent structure), and the blue line represents a signal together with stochastic or random noise. The fluctuating signal and random noise are too weak to induce a transition in the system individually. However, when the subthreshold fluctuating signal resonates or combines with a stochastic signal (i. e. stochastic resonance), the system can undergo a transition.

Due to internal noise (i.e., within a cell), the differing response to external cues can lead to variable outcomes, which are important to ensure the survival of at least some of the cell population (Maamar et al. 2007). Similarly, stochastic differences between cells in a tissue may be amplified to achieve different cell fates that are crucial in tissue or organ patterning (Meyer & Roeder, 2014). For example, in plants, experiments using a combination of genetics, cell biology and modelling have shown that noise is essential to achieve robust regulation of cell shape. For example, noise (in this case, heterogeneity in cell cycle timing, varying almost 3–5 fold between cells) is essential for proper sepal shape regulation (Hong et al., 2016). Interestingly, in sepals, mechanical feedback loops amplify differences between cells and contribute to sepal shape regulation (Hervieux et al., 2016). Thus, in addition to genomic mechanisms, non-genomic components play a crucial role in noise propagation.

A particularly interesting example of how noise is leveraged in the temporal regulation of developmental transitions involves noise acting as a facilitator in the seasonal alignment of seed germination. The timing of the decision of a dormant seed to germinate is one of the most consequential in the lifecycle of a plant (Finch-Savage & Leubner-Metzger, 2006). Whereas internal noise can facilitate bet-hedging, so that at least a few seeds undergo germination when environmental conditions are favourable (Pausas et al., 2022; Venable, 2007), the robust timing of seed germination instead involves leveraging external noise (Topham et al., 2017). In detailed work from the Bassel lab (Topham et al., 2017), it was shown that the robust timing of seed germination was modulated by noise in the spring, and noisy input (low temperature alternating with warm temperature) was more effective than continuous input with constant cold. Presumably, fluctuations in temperature are far more prevalent during seasonal transitions, and in this case, noise may serve as a harbinger of seasonal change to robustly time seed germination.

This observation raises the question as to why a system may leverage noise. Answering this question is not straightforward, but several explanations are possible. For instance, if germination required constant cold, this would limit dormancy breaking to a narrow window of time, making the system tight but inflexible. This could potentially delay germination because the seed would have to wait to break dormancy (thus acquiring competence until this condition was met). Moreover, year-to-year variations in the duration of optimal cold or when cold is experienced would affect the competence to germinate, and this could then impact post-

dormancy development and survival. Instead of using a variable input, i.e., noise, the seed can potentially bypass these problems and acquire competence to germinate at a time most favourable for starting growth after dormancy. Additionally, noise (daily temperature variation) is often more prominent in early spring, and thus, its use may be a way to recognise the advent of the spring-summer transition.

## Stochastic resonance, noise and time

The example of seed germination above demonstrates how noise can be leveraged for timing developmental transitions. Therefore, it is essential to consider the underlying mechanisms by which noise can be leveraged. In real-life situations, biological responses need to be robustly regulated spatially and/or temporally, and threshold-based switches play a crucial role in this (Sagner & Briscoe, 2017; Wolpert, 1996; (Cardelli et al., 2017). Whether spatial or temporal, a key problem faced by organisms is how to integrate noisy input and achieve robust regulation of such a threshold-based switch.

Stochastic resonance (SR) is a particularly attractive mechanism to consider in the context of noise leveraging associated with threshold-based switches, particularly in the temporal regulation of developmental transitions. It involves the cooperation of a low level of noise together with external forcing (or sub-optimal/sub-threshold levels) of an input signal to induce a switch or transition (McDonnell & Abbott, 2009; Wiesenfeld & Moss, 1995) (Figure 2). SR was first described to explain the periodicity of ice ages in the Earth's climate. Intriguingly, the occurrence of ice ages has a period of approximately 100000 years (i.e., an interglacial period of approximately 100000 years separating two consecutive ice ages) (Benzi et al., 1982). Early theories indicated that this periodicity was based on the orbital eccentricity of the Earth's rotation around the Sun, whose periodicity matched the period of glacial–interglacial climate change (Milankovic, 1998). However, later analysis (Benzi et al., 1982) showed that the orbital eccentricity could only account for a small amplitude of change in temperature. Instead, SR theory, which considered noise represented by a variable solar radiation output combined with the orbital eccentricity, was shown to accurately explain the 100000-year periodicity.

Since its introduction, SR has been applied to biological phenomena (McDonnell & Abbott, 2009; Wiesenfeld & Moss, 1995) (Figure 2). However, examples of SR in biological systems where the role of SR can be proven experimentally are scarce. Nevertheless, using simulations and experiments, the signature of SR has been

demonstrated in mechanoreceptors of hair cells of crayfish that respond to changes in water current to detect the presence of predators (Douglass et al., 1993). Since then, SR has been implicated in neuronal function as well (Faisal et al., 2008).

In plants, the counterintuitive phenomenon of SR has been used to explain the rapid gravitropic response (Meroz & Bastien, 2014). In gravitropism, the cascading of starch granules in response to a gravitropic stimulus in columella cells initiates a bending response (Haberlandt, 1900; Kiss & Sack, 1989). Inherent stochasticity (noise) by which starch granules fall to one side of the cell 'cooperates' with external forcing (gravitropic vector), which causes cascading of starch granules, can explain the kinetics of the gravitropic response. Noise, represented by the random/stochastic cascading of starch granules, on its own is too weak to initiate the bending, and pure forcing (without a low level of noise) by gravitropic stimuli is too slow to explain the rapid response. Only when the two combine (i.e., noise in phase with forcing) can a rapid gravitropic response occur on the timescales typically observed (Olivier Hamant, 2021).

As the latter example demonstrates, SR as a mechanism is particularly interesting when a temporal aspect of change is considered. For example, when a developmental transition is accomplished within a certain timeframe in response to an input signal that is noisy and could be subthreshold, one can imagine that due to subthreshold input, either this transition cannot be achieved in the relevant timeframe or if it would, then it could be after a delay beyond the required timeframe and thus have a negative impact on the organism. For example, in tree buds, experimental data have shown that while growth is optimal at ca. 20°C, in boreal regions, tree bud break occurs in early May when temperatures fluctuate, and mean temperatures are often far below those needed for optimal growth. However, if buds waited until temperatures were most optimal for growth, bud break would be more delayed, curtailing the growing season considerably. Although SR may not be involved in bud break, the above example highlights a potential advantage of the SR mechanism since sub-threshold inputs, when combined with noise (with the signal occasionally exceeding the threshold) could enable a developmental transition to proceed correctly within an appropriate timeframe, like in the above gravitropism example. Similarly, SR could explain the high sensitivity of sensory neurons to low sub-threshold signals despite high noise. It is this property of SR (combining a temporal response to low subthreshold signals combined with noise) that makes it an interesting mechanism to explore in biology. It is also possible that SR could be a key mechanism in adaptation that has not yet been recognised.

Intriguingly, in all the examples where SR has been invoked, there is also experimental evidence of a non-genomic mode of action, e.g., ion channels and the control of action potentials or the cascading of starch grains in plants. The question then is whether SR can be genetically encoded. It is plausible that SR could operate via genomic mechanism, e.g. at the level of transcription, at least theoretically. Nevertheless, at the moment, examples of this are lacking.

## Perspectives

As shown in the above examples, noise can be both detrimental and beneficial. Consequently, organisms have developed mechanisms to suppress noise and harness it. Whereas several mechanisms for noise suppression have been identified, less is known about mechanisms for leveraging noise. Understanding noise and its effects is also important in the context of impending climate change as incidents of environmental fluctuations are increasing, and deciphering noise suppression and noise leveraging mechanisms could provide insights into possible ways for climate change mitigation, e.g., by designing synthetic genetic circuits that enable leveraging noise. Extensive information has been gathered about the genes and genetic pathways essential for the functioning of cells or organisms. Such knowledge can now be used to study the impact of noise and address the complexity of cellular and organismal behaviour under natural conditions that are inherently noisy.

## Acknowledgements

The author (RPB) acknowledges Olivier Hamant for introducing the idea of noise and particularly the concept of stochastic resonance and for discussing on this review. Additionally, RPB would like to thank Prof. Malcolm Bennett, Prof. George Bassel, Dr. Kristoffer Jonsson for discussions on this review. RPB would like to thank Dr. Kristoffer Jonsson and Dr. Bibek Aryal for help with figures and formatting the manuscript. RPB is supported by grants from Swedish Natural Science Foundation (2020-03522), Human Frontiers Science Program (HFSP) RGP0002/2020.

**Conflict of interest.** The author (RPB) has no conflict of interest to declare.

**Data and coding availability statement.** This is a review and contains no data or coding that needs to be reported.

**Author contributions.** Rishikesh P. Bhalerao conceived and wrote the manuscript.

**Funding statement.** RPB is supported by grants from Swedish Natural Science Foundation (2020-03522) and Human Frontiers Science Program (HFSP) RGP0002/2020.

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
