## [Editor Report]

Thanks very much for this interesting submission on biological noise. The manuscript has now been assessed by two expert reviewers. They are positive about the manuscript but have some points about the framing, literature connections, and presentation that will be important to address going forward.

I have chosen “minor revision” because to my eyes the points raised throughout the reviews are somewhat easily addressable. However you will notice that one reviewer categorised their suggestions as “major”, and they indeed do include several mid-scale structural comments. Please do address these points carefully. Perhaps most importantly, the reviewer suggests several currently absent branches of the plant-specific literature which should be engaged with. Also, the relevance and positioning of this study for plant biology should be highlighted at higher level in the manuscript (title/abstract), and figures could be clarified as per the reviewer’s comments.

For my part, I was surprised by the absence of some classic work on noise in decision-making and information processing in more general contexts, e.g. Johan Paulsson’s work https://www.nature.com/articles/nature09333. Perhaps connecting with studies like this would strengthen the theoretical foundations of the review?

---

## [Editor Report]

Thanks for your responses to the reviewers' comments. Both now consider the manuscript scientifically complete and I am happy to agree. R2 suggests some styling and typo fixes; please take a look at these during production.